# High Glucose Treatment Limits Drosha Protein Expression and Alters AngiomiR Maturation in Microvascular Primary Endothelial Cells via an Mdm2-dependent Mechanism

**DOI:** 10.3390/cells10040742

**Published:** 2021-03-27

**Authors:** Brian Lam, Emmanuel Nwadozi, Tara L. Haas, Olivier Birot, Emilie Roudier

**Affiliations:** Muscle Health Research Center, Angiogenesis Research Group, Faculty of Health, School of Kinesiology and Health Science, York University, Keele Street, Toronto, ON 4700, Canada; brianl@yorku.ca (B.L.); enwadozi@yorku.ca (E.N.); thaas@yorku.ca (T.L.H.); birot@yorku.ca (O.B.)

**Keywords:** micro-RNA (miRs), MDM2, DROSHA, VEGF-A, hyperglycemia, angiogenesis, skeletal muscle, dermal, angiostatic, microvascular endothelial cells (MECs)

## Abstract

Diabetes promotes an angiostatic phenotype in the microvascular endothelium of skeletal muscle and skin. Angiogenesis-related microRNAs (angiomiRs) regulate angiogenesis through the translational repression of pro- and anti-angiogenic genes. The maturation of micro-RNA (miRs), including angiomiRs, requires the action of DROSHA and DICER proteins. While hyperglycemia modifies the expression of angiomiRs, it is unknown whether high glucose conditions alter the maturation process of angiomiRs in dermal and skeletal muscle microvascular endothelial cells (MECs). Compared to 5 mM of glucose, high glucose condition (30 mM, 6–24 h) decreased DROSHA protein expression, without changing *DROSHA* mRNA, *DICER* mRNA, or DICER protein in primary dermal MECs. Despite DROSHA decreasing, high glucose enhanced the maturation and expression of one angiomiR, miR-15a, and downregulated an miR-15a target: Vascular Endothelial Growth Factor-A (VEGF-A). The high glucose condition increased Murine Double Minute-2 (MDM2) expression and MDM2-binding to DROSHA. Inhibition of MDM2 prevented the effects evoked by high glucose on DROSHA protein and miR-15a maturation in dermal MECs. In *db/db* mice, blood glucose was negatively correlated with the expression of skeletal muscle DROSHA protein, and high glucose decreased DROSHA protein in skeletal muscle MECs. Altogether, our results suggest that high glucose reduces DROSHA protein and enhances the maturation of the angiostatic miR-15a through a mechanism that requires MDM2 activity.

## 1. Introduction

Angiogenesis is a dynamic process through which endothelial cells proliferate, migrate and assemble to form new and mature capillaries. This process expands the microvascular endothelium, providing an optimal surface to exchange nutrients and gases between blood and cells and also serving many physiological functions such as endocrine communication, immune protection and thermoregulation [1,2,3,4]. The different steps of the angiogenic process are well-orchestrated and tightly regulated by a plethora of pro-angiogenic and anti-angiogenic (angiostatic) signaling molecules [5,6]. The pro-angiogenic VEGF-A and the angiostatic Thrombospondin-1 (THBS-1) are among the most studied of these secreted matrix-bound and circulating proteins.

Micro-RNAs (miRs) are short RNAs that do not code for proteins but rather serve as a framework to translationally repress a set of genes [7]. AngiomiRs represent a subset of miRs that control the expression of pro-angiogenic and angiostatic genes, including *VEGF-A* and *THBS-1* [8]. One of the first miRs identified as a critical regulator of angiogenesis is miR-126, the most abundant miR in endothelial cells [9,10,11,12]. Targeted deletion of miR-126 in mice causes leaky vessels, hemorrhage and embryonic lethality [12]. miR-126 can exert a pro-angiogenic effect by targeting the Sprouty-related protein SPRED-1, a negative regulator of VEGF-A signaling [10,12]. By downregulating SPRED-1 expression, miR-126 helps microvascular endothelial cells (MECs) sense VEGF-A signal and thus facilitates angiogenesis [12]. The polycistronic miR-17–92 cluster encodes for 7 mature miRs (miR-17-3p, miR-17-5p, miR-18a, miR-19a, miR-20a, miR-19b and miR-92a-1) that regulate the expression of extracellular matrix proteins [13,14]. Interestingly, in this cluster, miR-18a and miR-19a repress the expression of the angiostatic THBS-1 and therefore enhance the angiogenic potential of microvascular endothelial cells [14,15]. Conversely, some angiomiRs present angiostatic properties. For example, miR-15a can repress the expression of VEGF-A, its receptor VEGF-R2 and the endothelial-specific receptor tyrosine kinase Tie 2 [16,17], promoting angiostasis in MECs [18].

Diabetes is associated with functional and structural microvascular alterations in the skeletal muscle and cutaneous tissues [19,20,21,22,23]. The skin microvasculature presents a lower capillary density and a decreased capacity to vasodilate in diabetic patients [24,25,26]. Hyperglycemia is correlated with capillary rarefaction in the skeletal muscle of type 1-diabetic rats [27,28,29]. Whether type-2 diabetes induces a loss of capillaries in skeletal muscles remains unclear as different animal models and human studies have reported conflicting data [22,30,31,32]. Nevertheless, alterations of the skeletal muscle microvascular structure correlate with the severity of type 2-diabetes [32,33,34,35]. These observations indicate that diabetes impairs the angiogenic process in these tissues, and indeed, diabetes significantly alters the expression of pro-angiogenic and angiostatic factors in the skeletal muscle and cutaneous tissues [22,36,37].

We have previously reported that the activity of the E3 ubiquitin ligase Murine Double Minute 2 (MDM2) supports skeletal muscle angiogenesis and the outgrowth of MECs from skeletal muscle explants ex vivo [23,38]. Using dermal MEC, we have shown that the constitutive activation of MDM2 decreases the expression of *THBS1* by 46% and *VEGF-A* by 20% and enhanced the migratory capacity of MECs [38]. Altogether, our previous results illustrate that MDM2 can modulate the angiogenic capacity of dermal and skeletal muscle MECs by controlling the expression of pro-angiogenic and angiostatic genes. Our group and others have reported that chronic hyperglycemia was associated with an upregulation of angiostatic molecules in human primary dermal MECs and rodent skeletal muscle MECs, reducing their migratory capacity [37,39,40,41,42]. However, whether a direct link exits between high glucose exposure and MDM2 ability to regulate the expression of pro-angiogenic and angiostatic genes in MECs remains unknown.

Originally, angiomiRs were discovered when loss-of-function experiments reported a decreased angiogenic capacity in MECs when genes coding for two enzymes involved in miRs maturation, the ribonuclease DROSHA and the endoribonuclease DICER, were knocked-down [43,44,45]. The canonical miR biogenesis starts with the transcription of a primary miR (pri-miR) transcript. In the nucleus, DROSHA binds to DGCR8 to form the microprocessor complex that identifies and cleaves long primary miRs (pri-miRs) into precursor miRs (pre-miRs) [46]. Pre-miRs are around 65-nucleotides long and present a stem-loop structure. Once exported into the cytoplasm, pre-miRs are processed into their mature forms (miR) by DICER, which removes the stem-loop from pre-miRs [47]. This last step generates mature miRs of about 22 nucleotides [7].

While angiomiRs repress transcripts of pro-angiogenic and angiostatic genes in MECs to control functional and structural microvascular adaptations [8,48,49], emerging evidence suggests that high glucose conditions alter the expression levels of angiomiR in endothelial cells [50,51,52,53]. Nevertheless, it is unknown whether such high glucose conditions can alter the molecular process of angiomiR biogenesis in dermal and skeletal muscle microvascular endothelial cells. Recent reports indicate that MDM2 senses nutrient stress in tumor cells and mouse embryonic fibroblasts [54,55], altering the process of miR maturation in these cells [54]. Based on our knowledge that MDM2 regulates the expression of pro-angiogenic and angiostatic factors in MECs [39], we postulate that high glucose conditions might alter the angiomiRs biogenesis process through a MDM2-dependent mechanism, controlling the expression of angiogenesis-related genes in MECs.

## 2. Materials and Methods

### 2.1. Mouse Model

All experiments were conducted according to the Canadian Council on Animal Care and with the approval of the York University Animal Care Committee (committee approval number: 2017-19 and 2107-20). Male homozygous leptin-resistant mice on a C57BL/6J background (B6.BKS(D)-Leprdb/J; referred to as *db/db*) and age-matched wild-type C57BL/6J mice (referred to as *wt/wt*) were purchased from The Jackson Laboratories (Bar Harbor, ME, USA) and used for experiments at 13 weeks of age. Breeding of heterozygous *db/wt* mice, also purchased from Jackson Laboratories, generated additional male mice that were assessed at 4 weeks (2 *wt/wt*, 2 *db/db* and 9 *db/wt* mice). In these 4-week aged mice, blood glucose measurements were assessed by saphenous vein blood and measurement using a glucometer, ranging from 5.8 to 20.5 mM of glucose (Aviva Accu-Chek; Roche Diagnostics, Indianapolis, IN, USA). For both age groups, gastrocnemius muscles were collected and immediately frozen in liquid nitrogen until analysis, as previously reported in [56].

### 2.2. Primary Microvascular Endothelial Cell Cultures

All primary microvascular endothelial cells were used for experiments at passages 6 to 8 and cultured on rat tail type-1 collagen-coated culture dishes (50 µg/mL in 0.02 M acetic acid: Gibco, cat. no. A10438-01, Thermofisher, Burlington, ON, Canada). Primary human dermal microvascular endothelial cells (HDMECs, cat. no. 2000) were purchased from ScienCell Research Laboratories (Carlsbad, CA, USA) and cultured as previously described [39]. Briefly, cells were cultured in ECM (cat. no. 1001) supplemented with 5% FBS (cat. no. 0025), 1% EC growth supplement (ECGS, cat. no. 1052) and an antibiotic solution containing 100 U/mL penicillin and 100 mg/mL streptomycin (cat. no. 0503); all from ScienCell Research Laboratories (Carlsbad, CA, USA). Primary mouse skeletal muscle-derived endothelial cells (mSMECs) were isolated in our laboratory as previously described [57]. mSMECs were cultured in low glucose DMEM (Gibco cat. no. 11885092, Thermofisher, Burlington, ON, Canada) supplemented with 12% FBS (cat. no. 080150; Wisent, St Bruno, QC, Canada) and an antibiotic solution containing 100 U/mL penicillin and 100 mg/mL streptomycin (Gibco, cat. no. 15140122; Thermofisher, Burlington, ON, Canada).

### 2.3. D-Glucose and D-Mannitol Cell Treatment

Cells were respectively plated on 6-well dishes (125,000 cells/well) or 60-mm cell culture dishes (300,000 cells/dish) for RNA or protein isolation. Upon reaching 80% cell confluence, 10% D-glucose solution (cat. no. 47829; Sigma-Aldrich, Oakville, ON, Canada) or 10% D-mannitol solution (cat. no. M4125-100G; Sigma-Aldrich, Oakville, ON, Canada) in ddH2O was added to the cell cultures to reach final glucose/mannitol concentrations of 5, 10, 20, 30 or 40 mM. Cells were incubated for 0.5, 1, 2, 3, 6 and 24 h at 37 °C and 5% CO_2_ before harvesting.

### 2.4. Inhibition of MDM2 Activity

HDMECs were incubated with 10 µM of following inhibitors for one hour prior to glucose treatment: Nutlin-3 (cat. no. N6287-5MG; Sigma-Aldrich, Oakville, ON, Canada), or MX69 (cat. no. S8403; Selleckchem, Houston, TX, USA), or RG7112 (cat. no. S7030; Selleckchem, Houston, TX, USA). Cells were subsequently treated with 5- or 30-mM glucose for 6 h and 24 h before cell harvesting.

### 2.5. Immunoblotting

Immunoblotting was carried out on protein extracts from primary endothelial cells and mouse gastrocnemius muscles as previously described [23,38]. Briefly, muscle samples were homogenized in a protein lysis buffer (ratio weight:volume of 1:15) composed of 50 mM Tris-base, 100 mM NaCl, 5 mM EDTA, 1% sodium deoxycholate, 1% triton X-100, 1 mM phenylmethylsulfonyl fluoride (PMSF), 1 mM NaF, 1 mM Na3VO4, protease inhibitor cocktail (Roche Complete Mini, cat. no. 04906845001; Sigma-Aldrich, Oakville, ON, Canada), phosphatase inhibitor cocktail (Roche PhosSTOP cat. no. 11836153001; Sigma-Aldrich, Oakville, ON, Canada), pH 8. Cells were harvested using the same lysis buffer (150 µL for 800,000 cells). After 20 min incubation at 4 °C, homogenates were centrifuged at 16,000 g for 15 min, and supernatants were collected. After total protein concentration determination (BCA assay, cat. no. B9643; Sigma-Aldrich, Oakville, ON, Canada), denatured sampled (10 to 20 µg of total proteins per well) were separated by SDS-PAGE and blotted onto nitrocellulose membranes (Whatman BA95; Sigma-Aldrich, Oakville, ON, Canada). A calibrator (i.e., loading of a protein extract sample composed of a pool of all samples) was loaded on each gel to conduct inter-gel comparisons. Quality of the transfer was controlled by Ponceau S Red staining. After blocking with 5% fat-free milk (dissolved in tris-buffered saline with 0.1% TWEEN 20), the blots were probed with appropriate primary antibodies. The following primary antibodies were used: a non-commercial anti-Murine Double Minute-2/Human Double Minute-2 (MDM2/HDM2; supernatant from the hybridoma (clone 2A10) previously described [58] and kindly provided by Dr. Mary Ellen Perry; National Cancer Institute, Frederick, MD, USA, see [23]), anti-MDM2/HDM2 (Santa Cruz Biotechnologies Santa-Cruz, CA, USA, cat. no. sc-965), anti-VEGF-A (VG-1, Milllipore, Missisauga, ON, Canada, cat. no. 05-1117), anti-DROSHA, anti-DICER (Cell Signaling Technology, New England Biolabs, Whitby, ON, Canada, cat. no. 3364 and 5362, respectively), anti-thrombospondin-1 (THBS-1 clone A6.1; Invitrogen, Thermofisher, Burlington, ON, Canada, cat. no. MA5-13398). α/β-TUBULIN was detected as a loading control (Cell Signaling Technology #2418S, New England Biolabs, Whitby, ON, Canada). After incubation with the appropriate HRP-conjugated secondary antibody, either anti-rabbit and light chain specific anti-mouse (Cell Signaling Technology, cat. no. 7074 & cat. no. 55802, respectively) or anti-mouse (DAKO, Burlington, ON, Canada; cat. no. P0260) proteins were visualized with enhanced chemiluminescence (Pierce ECL, cat. no. 32106; Thermofisher, Burlington, ON, Canada) on an imaging station (Kodak 4000 MM Pro; Carestream Health, Concord, ON, Canada) or on X-ray film (CL-XPosure, cat. no. 34090; Thermofisher, Burlington, ON, Canada). Blots were analyzed with Carestream Molecular Imaging software.

### 2.6. Protein Co-Immunoprecipitation Assay

HDMECs were cultured with either 5 or 30 mM glucose for 6 h before cell lysis. Cells were also treated with 20 µM MG-132 (EMD Millipore, cat. no. 474790; Sigma-Aldrich, Oakville, ON, Canada) in order to block proteasomal degradation. After 30 min pre-clearing of the cell lysates with normal mouse IgG, cell lysates were incubated overnight with agarose-conjugated anti-DROSHA or agarose-conjugated normal mouse IgG (Santa Cruz Biotechnology, Santa-Cruz, CA, USA cat. no. sc-393591AC & sc-2343; respectively) for immunoprecipitation. Supernatants were denatured and run on SDS-PAGE. DROSHA and MDM2 protein expression levels were detected by immunoblotting as described above using the following antibodies: Anti-DROSHA (cat. no. 3364; Cell Signaling Technology, New England Biolabs, Whitby, ON, Canada), anti-MDM2 (non-commercial clone 2A10 or cat. no. sc-965 from Santa Cruz Biotechnology). Anti-rabbit and light chain specific anti-mouse (Cell Signaling Technology, cat. no. 7074 & 55802, respectively) were used to detect DROSHA and MDM2, respectively.

### 2.7. Isolation of pri-miR, miR, and total RNA. RT-qPCR

Total RNAs were isolated from cell lysates using QIAzol Lysis Reagent (Qiagen, Toronto, ON, Canada, cat. no. 79306) for experiments where mRNA levels alone were assessed. For experiments combining mRNA, pri-miR and miR measurements from the same samples, total RNAs were purified using the miRNeasy Micro Kit (Qiagen, cat. no. 217004) to ensure the efficient yield of RNAs from 18 nucleotides to upwards. A thousand fmol of cel-miR-39-3p was added to each RNA isolate as an exogenous control to measure mature miR. RNA purity and concentration were determined by measuring absorbance at 260 and 280 nm. High-Capacity RNA-to-cDNA Kit (Applied Biosystems, cat. no. 4387406; Thermofisher, Burlington, ON, Canada) was used to reverse transcribe (RT) mRNA and pri-miR into cDNAs, while the Taqman^®^ Advanced miRNA cDNA Synthesis Kit (Applied Biosystems, cat. no. A28007; Thermofisher, Burlington, ON, Canada) was used to reverse transcribe miR. Semi-quantitative real-time PCR was conducted for each sample in triplicate using TaqMan^®^ Gene Expression, TaqMan^®^ MicroRNA or TaqMan^®^ Advanced MicroRNA Assays and a TaqMan^®^ Universal Master Mix II for 40 cycles of denaturation at 95 °C for 3 s, and annealing and extension at 60 °C for 30 s. *DROSHA, DICER1, THBS-1, VEGF-A, MDM2,* pri-miR-126, pri-miR-17-92, pri-miR-15, miR-126-3p, miR-126-5p, miR-18a, and miR-15a were measured (list of probes used are indicated in Table 1). The ∆∆Ct method was used to express the relative changes. mRNA and pri-miR transcripts were normalized to Hypoxanthine-guanine phosphoribosyltransferase (*HPRT*) gene (for) and mature miR to cel-miR-39-3p. Data is expressed as $2^{-\Delta \Delta Ct}$.

### 2.8. Statistical Analysis

Statistical analyses were performed with Student’s *t* test and 1- and 2-way ANOVAs with Prism8 (GraphPad, San Diego, CA, USA). For 1- and 2-way ANOVAs, Sidak’s multiple comparison and Bonferroni post hoc tests were used, respectively. Correlation analyses between variables were performed with nonparametric 2-tailed Pearson correlation with determination of Pearson r and r2. *p* < 0.05 was considered to be statistically significant.

## 3. Results

### 3.1. High Glucose Treatment Alters the miR Biogenesis Machinery by Decreasing DROSHA Protein Expression in Human Primary Dermal Microvascular Endothelial Cells

We evaluated here whether 24 h of high glucose treatment (30 mM) alters DROSHA and DICER expression in primary human dermal MECs (HDMECs). High glucose treatment significantly decreased DROSHA protein expression by 31% compared to normal glucose control conditions (5 mM, *p* = 0.0004, Figure 1A). High glucose treatment (30 mM) had no effect on DICER protein levels (Figure 1B) and did not affect DROSHA and DICER1 mRNA levels (Figure 1C,D). This suggested that high glucose conditions might regulate DROSHA expression through a post-transcriptional mechanism.

### 3.2. High Glucose-Mediated Decrease in DROSHA Protein Expression Is Dose- and Time-Dependent but Is Not Affected by Osmotic Stress in Primary Microvascular Endothelial Cells

#### 3.2.1. Dose-Effect of Glucose Treatment on DROSHA and DICER Protein Expression

To further characterize the effect of high glucose treatment on endothelial DROSHA protein levels, and to confirm the absence of effect on DICER, a glucose dose-response of protein expression was assessed by treating HDMEC for 24 h with increasing glucose concentrations (5 to 40 mM, Figure 2A,B, *p* = 0.017). DROSHA protein expression decreased by 20% at 30 mM and 23% at 40 mM compared to 5 mM glucose condition (Figure 2A, 0.80 ± 0.04 and 0.77 ± 0.07 versus 1.00 ± 0.05, *p* < 0.05). In line with Figure 1, no change in DICER protein expression was observed with increasing glucose concentrations (Figure 2B).

#### 3.2.2. Time-Effect of High Glucose Treatment on DROSHA Protein Expression

A time effect of high glucose treatment (30 mM) on endothelial DROSHA protein expression was observed (Figure 2C, *n* = 3, 2W-ANOVA, time-effect *p* = 0.025, glucose effect *p* = 0.0003). Whereas 3 h incubation with 30 mM glucose did not significantly change DROSHA protein expression (0.66 ± 0.01 at 30 mM vs. 0.73 ± 0.02 at 5 mM, *p* = 0.57), the effect of high glucose became significant after 6 h incubation (−19%, 0.58 ± 0.03 at 30 mM vs. 0.72 ± 0.05 at 5 mM, *p* < 0.05) and was confirmed at 24 h in line (−33%, 0.47 ± 0.06 at 30 mM vs. 0.70 ± 0.03 at 5 mM, *p* < 0.001).

#### 3.2.3. Mannitol Treatment Does Not Affect DROSHA Protein Expression

The incubation of HDMECs with 30 mM mannitol concentration for 24 h did not alter DROSHA protein expression (Figure 2D) suggesting that the observed effect of high glucose treatment on endothelial DROSHA protein expression was independent of any increase in the osmotic pressure.

### 3.3. High Glucose Treatment Shifts Endothelial-Derived Mature angiomiR Expression towards an Anti-Angiogenic Profile without Altering Primary pri-miR Expression

We next wanted to test whether the maturation process of angiomiRs was altered in HDMECs. We analyzed the impact of high glucose treatment on the biogenesis of angiomiRs miR-126, miR-18a and miR-15a. These angiomiRs were selected as they are expressed in MECs [12,16,59] and are known regulators of the expression or signaling of THBS-1 or VEGF-A, respectively two of the most well-characterized angiostatic and pro-angiogenic factors [14,16,18,59].

#### 3.3.1. High Glucose Treatment Did Not Alter the Expression of pri-angiomiRs pri-miR-126, pri-miR-15 and pri-miR-17-92 in Human Primary Dermal Microvascular Cells

The *Mir126, Mir15a/16-1, Mir17-92* loci produce the following primary miR (pri-miR) transcripts: pri-miR-126, pri-miR-15a/16-1 (referred in Figure 3 as pri-miR-15a) and pri-miR-17-92, respectively. After processing by DROSHA and DICER, pri-miR-126 gives rise to two mature pro-angiogenic miRs (miR-126-3p and miR-126-5p); pri-miR-15a/16-1 produces the angiostatic miR-15a and miR-16-1, and the pri-miR-17-92 is cleaved to generate seven different mature miRs, including the pro-angiogenic miR-18a. As shown in Figure 3A, high glucose treatment had no significant effect on pri-miR expression levels in HDMECs.

#### 3.3.2. High Glucose Treatment Upregulated the Expression of the Angiostatic Mature miR-15a without Altering the Expression of the Pro-Angiogenic Mature miR-126-3p, miR-126-5p, and miR-18a

Both miR-126-3p and miR-126-5p can exert pro-angiogenic effects via the reinforcement of the pro-angiogenic VEGF-A signaling [12,59]. The treatment of HDMEC for 24 h with high glucose concentration (30 mM) did not change the levels of the mature miR-126-3p and miR-126-5p (Figure 3B, *n* = 9–10). The pro-angiogenic miR-18a, whose predicted target genes (downregulation) include the anti-angiogenic thrombospondin-1 (THBS-1) was not affected by the high glucose treatment (Figure 3B: 0.81 ± 0.11 at 30 mM vs. 1.02 ± 0.09 at 5 mM). Conversely, miR-15a can exert some angiostatic effects by down-regulating VEGF-A expression [18]. High glucose concentration (30 mM) increased miR-15a expression by +198% compared to low glucose concentration (5 mM) (Figure 3B: 2.7 ± 0.52 at 30 mM vs. 0.90 ± 0.02 at 5 mM, *p* = 0.0007).

#### 3.3.3. High Glucose Treatment Results in Decreased VEGF-A Protein Expression but No Change in THBS-1 Protein Levels

Next, we assessed the expression levels of *VEGF-A* and *THBS-1* mRNA and proteins in glucose-treated HDMECs (Figure 3C–F). Exposure to high glucose concentrations did not induce a significant change in *VEGF-A* (*p* = 0.689) or *THBS-1* mRNA (*p* = 0.156, Figure 3C,D). However, in line with the increase in mature miR-15a (Figure 3A), VEGF-A protein expression level was significantly reduced by approximately 53% (Figure 3E, 0.04 ± 0.01 at 30 mM vs. 0.08 ± 0.01 at 5 mM, *p* < 0.05). THBS-1 protein levels were not affected (Figure 3D 1.17 ± 0.08 at 30 mM vs. 1.00 ± 0.02 at 5 mM, *p* = 0.07) in line with the absence of change in mature miR-18a (Figure 3F).

### 3.4. Reduced Endothelial DROSHA Expression in Response to High Glucose Treatment Is Regulated by Murine Double Minute-2/DROSHA Interaction and MDM2 Activity

#### 3.4.1. High Glucose Treatment Increased MDM2 mRNA and Protein Expression in Human Primary Dermal Microvascular Endothelial Cells

MDM2 has been shown to interact with DROSHA [54]. To test the hypothesis that MDM2 could regulate the high glucose-mediated decrease in DROSHA protein expression in our conditions, we measured the expression levels of *MDM2* mRNA and protein in HDMECs treated for 24 h with low (5 mM) or high (30 mM) glucose (Figure 4A,B). Both mRNA and protein levels were significantly increased in response to high glucose treatment by +86% and +49%, respectively (Figure 4A, mRNA: 1.87 ± 0.33 at 30 mM vs. 1.00 ± 0.87 at 5 mM, *p* = 0.02; Protein: 1.49 ± 0.1 at 30 mM vs. 1.0 ± 0.07 at 5 mM, *p* < 0.001).

#### 3.4.2. High Glucose Treatment Enhanced MDM2 Binding to DROSHA in HDMECs

We next examined whether high glucose treatment could modulate the ability of MDM2 to bind to DROSHA in primary dermal MECs. As MDM2 undergoes proteasomal degradation with its targets [60], MECs were treated with MG-132, an inhibitor of the proteasome before immunoprecipitation assays were performed. As shown in Figure 4C, the ratio between co-immunoprecipitated MDM2 and DROSHA proteins was strongly increased after 6 h of treatment at 30 mM of glucose. The MDM2:DROSHA ratio showed a fold of change of 6.1 ± 1.6 at 30 mM compared to 5 mM of glucose (*p* < 0.05), showing that MDM2 and DROSHA interaction is promoted by high glucose treatment.

#### 3.4.3. Inhibition of MDM2 Activity Prevented the Decrease in DROSHA Protein Levels

We next examined whether MDM2 is required to decrease DROSHA protein expression under high glucose conditions. We analyzed the effect of high glucose concentration (30 mM, 6 h) on DROSHA protein in HDMECs pretreated with chemical inhibitors of MDM2; MX69, Nutlin-3 and RG7112 (1 h, Figure 4D).

MX69 binds to MDM2 RING domain, promoting MDM2 self-ubiquitination and degradation [61]. RG7112 and Nutlin-3 bind to the hydrophobic pocket of MDM2, restraining MDM2 capacity to bind to its canonical targets, which reduces MDM2 self-degradation [62]. Independently of glucose concentration (MDM2 and DROSHA in HDMECs treated with 30 or 5 mM glucose and MDM2 inhibitors (No inhibitors, MX69, Nutlin-3 and RG7112) for 6H. α,β-TUBULIN was used as a loading control. Quantification shows means ± SEM, *n* = 6 (Appendix A). MX69 treatment reduced MDM2 protein level in HDMECs from 0.43 ± 0.04 to 0.28 ± 0.03 (mean ± SEM, *p* < 0.01, untreated vs. MX69 treated, *t*-test, *n* = 12); while Nutlin-3 and RG7112 increased MDM2 protein levels (0.69 ± 0.05 and 1.40 ± 0.1 vs. 0.43 ± 0.04, Nutlin-3 and RG7112 vs. untreated, *p* ≤ 0.0001, *t*-test, *n* = 12).

High glucose treatment resulted in a significant decrease in DROSHA protein expression in the untreated HDMECs only (0.67 ± 0.04 vs. 0.42 ± 0.07, 5 mM vs. 30 mM). In HDMECs pretreated with the inhibitors, DROSHA protein remained unchanged by the high glucose condition (MX69: 0.66 ± 0.06 vs. 0.55 ± 0.06, Nutlin-3: 0.73 ± 0.07 vs. 0.80 ± 0.07, RG7112: 0.73 ± 0.06 vs. 0.70 ± 0.05; 5 mM vs. 30 mM, all non-significant). There was an overall effect of MDM2 inhibition on DROSHA protein level (*p* = 0.0051, Figure 4D). After 6 H of high glucose, HDMECs pretreated with Nutlin-3 and RG7112 presented higher levels of DROSHA protein compared to untreated HDMECs (0.80 ± 0.07 and 0.70 ± 0.05 vs. 0.42 ± 0.07; Nutlin-3 and RG7112 vs. untreated). This was not the case for HDMECs pretreated with MX69.

#### 3.4.4. Inhibition of MDM2 Activity by Nutlin-3 Prevented the Increase in the Mature Form of miR-15a

To assess the effect of MDM2 inhibition on the maturation of miRs, we measured the expression of angiomiR in HDMECs pretreated with Nutlin-3 and exposed to 5 and 30 mM of glucose for 24 H. Glucose and Nutlin-3 had no effect on the expression of miR-126-3p and miR-18a. Nutlin-3 and glucose had an overall impact on miR-15a expression (*p* = 0.03 for Nutlin-3 and *p* = 0.025 for glucose). We confirmed that 30 mM increased the expression of miR-15a. In the presence of Nutlin-3, no increase in miR-15a expression was observed in HDMECs despite high glucose treatment. This suggests that inhibition of MDM2 by Nutlin-3 prevented the upregulation of the mature form of miR-15a (3.7 ± 1.1 vs. 1.1 ± 0.2, untreated vs. Nutlin-3 treated, at 30 mM of glucose, *p* = 0.008, Figure 4E).

### 3.5. DROSHA Protein Expression Is Reduced in High Glucose-Treated Murine Primary Skeletal Muscle Endothelial Cells and In Vivo in Skeletal Muscles from Diabetic Mice

We investigated if high glucose conditions could also alter DROSHA expression in the skeletal muscle, a tissue also known to display high glucose-mediated endothelial dysfunction and impaired angiogenic activity. We observed that high glucose treatment decreased DROSHA protein expression in mouse primary microvascular endothelial cells isolated from skeletal muscle (mSMECs, Figure 5A, 0.78 ± 0.05 vs. 0.44 ± 0.01, 5 mM vs. 30 mM, 24 h, *p* < 0.05). To further investigate whether glucose concentration could regulate DROSHA expression in vivo, we measured DROSHA protein levels in gastrocnemius muscles from adults (13 weeks old) wild-type (*wt/wt*) and diabetic (*db/db*) mice (Figure 5B). DROSHA protein levels were significantly reduced in diabetic skeletal muscles by about 31% (0.36 ± 0.03 in *db/db* vs. 0.53 ± 0.05 in *wt/wt*, *p* < 0.05), while miR-15a expression and MDM2 protein levels remained unchanged (Appendix A).

Based on our previous data indicating that DROSHA decreased expression in HDMEC follows glucose dose-response, we selected animals expressing the intermediate phenotype *wt/db* at a younger age (4 weeks old) when metabolic disturbances are still under development. This allowed us to compare the protein expression levels of DROSHA in gastrocnemius muscles with corresponding resting blood glucose levels. As shown in Figure 5C, a negative correlation was observed between DROSHA protein levels and resting blood glucose (slope = −0.05, r^2^ = 0.45, *p* = 0.006).

## 4. Discussion

Oscillations in plasma glucose are an important consideration when managing diabetes and have been reported to cause microvascular endothelial alterations [63,64]. Under hyperglycemic conditions, endothelial cells have been described to become dysfunctional [65]. AngiomiRs represent a family of micro-RNA with pro- or anti-angiogenic properties that can modulate endothelial cell function [8]. The biogenesis of miRs is a complex process comprised of the transcription of primary miRs (pri-miR) from DNA, followed by a multi-step process of maturation that produces mature miRs. Changes in miRs expression levels in response to hyperglycemia were found to be involved in the development of endothelial dysfunction [66,67]. Nevertheless, the impact of high glucose conditions on the biogenesis process (transcription and maturation) of miRs, and particularly angiomiRs, remains largely unknown.

Here, we report that high glucose alters the process of miR maturation by downregulating DROSHA protein in primary skeletal muscle and dermal MECs. In vitro, high glucose increased the maturation of miR-15a specifically, while the maturation of other angiomiRs (miR-126a-3p, miR-126a-5p, and miR-18a) remained unchanged in primary MECs. These changes appeared to be MDM2-dependent. Indeed, high glucose increased MDM2-binding to DROSHA, and MDM2 inhibition prevented both the downregulation of DROSHA and the increase in miR-15a expression induced by high glucose.

In this study, high glucose conditions promoted a time- and dose-dependent decrease in DROSHA protein in primary skeletal muscle and dermal MECs and in the skeletal muscle tissue. The decrease in DROSHA protein induced by high glucose could not be explained by an increased transcription of DROSHA gene. Indeed, *DROSHA* mRNA remained unchanged by high glucose conditions. High glucose conditions appeared to downregulate DROSHA through posttranscriptional mechanisms. The canonical function of DROSHA is controlled through binding to other proteins. This is the case when p53, MDM2 or BRCA1 binds to the microprocessor complex [54,68,69]. Here, we report an increased interaction between MDM2 and DROSHA under high glucose conditions. As an E3 ubiquitin ligase, MDM2 has been reported to modulate the stability and function of many of its targets by direct binding and through ubiquitination [60]. MDM2 was recently identified in vitro as a binding partner and an E3 ubiquitin ligase for DROSHA in ubiquitination assays [55]. In the same study, Ye et al. observed that silencing of MDM2 reduces DROSHA ubiquitination in HEK293 cancer cells [54]. The author noticed a reduced proteasomal accumulation of MDM2 in A459 lung tumor cells subjected to glucose deprivation (glucose-free media) that was associated with a concomitant increase in DROSHA protein levels [54]. This supports the notion that glucose deprivation restrains MDM2′s capacity to downregulate DROSHA through ubiquitination. Taking in consideration their data and our present observation that MDM2 inhibition prevented the downregulation of DROSHA induced by high glucose, our work suggests that high glucose enhances the capacity of MDM2 to downregulate DROSHA. In our experiments, RG7112 and Nutlin-3 inhibitors were more efficient in preventing DROSHA downregulation than MX69, potentially indicating that the hydrophobic pocket of MDM2 could support the DROSHA-MDM2 interaction under high glucose condition. Altogether, for the first time in a non-cancer context, we report that high glucose downregulates DROSHA in an MDM2-dependent manner.

To our knowledge, only one study has investigated the impact of glucose variations on DROSHA expression in endothelial cells. The authors indicated that a high glucose concentration (25 mM) increased mRNA expression levels of *DROSHA*, *DGCR8*, *DICER* and *AGO-2* in human umbilical vein endothelial cells (HUVECs) [70]. Interestingly, they described increased *DROSHA* mRNA expression when we observed decreased DROSHA protein levels. Unfortunately, they did not measure DROSHA or DICER protein levels. Moreover, HUVECs are macrovascular cells of venous origin and present major phenotypical differences when compared to microvascular ECs [71]. This could explain the apparent discrepancy between their mRNA observation and our DROSHA protein alterations in dermal and muscle primary microvascular endothelial cells and in vivo muscle tissue.

Despite a robust decrease in DROSHA protein expression, an important player of the miR maturation process, high glucose concentrations did not decrease the expression levels of the few selected angiomiRs we measured. In fact, high glucose concentration increased the expression of one of them, the mature anti-angiogenic miR-15a in MECs. Ye et al. showed that glucose deprivation stabilized DROSHA and enhanced the abundance of miRs (miR-376, miR-567 and miR-627-5p) known to regulate cell survival in tumoral A549 cells [54]. It is important to note that these authors did not analyze angiomiR and did not use endothelial cells. This raises the questions of whether DROSHA function could be regulated in a cell-type specific manner (i.e., endothelial-specific) or whether pri-angiomiRs could be matured in a miR-specific manner due to their specific abundance or sequence.

Beyond its role in miR processing, DROSHA also has non-canonical functions that include mRNA destabilization, transcription regulation, alternative splicing of exons, or even the maintenance of the genome integrity [72,73,74]. These non-canonical functions of DROSHA compete with it its main role of cleaving pri-miRs into pre-miRs [73,74]. We could then speculate that as DROSHA protein decreases under high glucose conditions, DROSHA gets confined to its primary role of pri-miR cleavage to maintain miR biogenesis to the detriment of its non-canonical functions. As a result, DROSHA function specifically in pri-miR maturation would be preserved, explaining the absence of a decrease in miRs levels. Interestingly, a recent report by Pendzialek and colleagues indicated that maternal diabetes promotes the exclusion of DROSHA from the nuclei of trophoblast cells [75]. In these cells, with reduced expression of nuclear DROSHA, the ability of DROSHA to maintain its miR maturation function seems to vary depending on the relative abundance of pri-miRs and the capacity of DROSHA to bind to them [75].

DROSHA-independent mechanisms might be involved in the upregulation of miR-15a in microvascular endothelial cells under high glucose condition. The upregulation of *MDM2* mRNA with high glucose suggests a putative upregulation of P53 activity. Since P53 enhances the post-transcriptional regulation of miR [68,76], the idea that the maturation of angio-miR could be regulated by p53 is appealing. Our results point out that future studies investigating the crosstalk between DROSHA, P53 and MDM2 could bring new insight regarding the underlying mechanisms of miR-15a maturation in MECs when glucose varies.

In the context of diabetes, uncontrolled and excessive angiogenesis in the retina can lead to diabetic retinopathy and blindness, while a reduced angiogenic capacity will contribute to altered skin and skeletal muscle function [22,25,30,32,39,77]. The expression of miR-15a was reported to be decreased in human retinal endothelial cells isolated from diabetic patients, supporting the notion that miR-15a has angiostatic functions [78], probably through its capacity to repress VEGF-A expression [17]. Here, we observed that high glucose increases miR-15a maturation and abundance in MECs from dermal tissues where angiogenesis capacities are restrained during diabetes.

Our data also questions whether MDM2 can downregulate VEGF-A through a DROSHA/*miR-15a* mechanism. The enhanced abundance of miR-15a observed in dermal MECs was associated with an important decrease in VEGF-A protein expression despite no change in *VEGF-A* mRNA level. The expression of miR-15a has previously been reported to reduced VEGF-A protein levels in circulating early precursors of endothelial cells without any change at the mRNA level [79]. miRs have the capacity to reduce gene expression through multiple ways involving RNA destabilization but also through the repression of protein translation [7]. While MDM2 can bind numerous proteins, it also has RNA binding domains and can interact with specific RNAs under specific contexts such as translational stress [80,81]. It remains unknown whether MDM2′s capacity to bind to RNAs could support the effects observed here. Future investigations will be necessary to delineate whether MDM2 could interact with pri-miR-15a/16-1 to favor its cleavage by DROSHA. Finally, MDM2 was reported to interact with the 3′UTR region of *VEGF-A* mRNA in tumoral cells [82]. One might wonder whether MDM2 binding to *VEGF-A* mRNA could promote a stronger interaction between miR-15a and the *VEGF-A* transcript and thus a greater repression of VEGF-A protein translation in MECs exposed to high glucose.

Here, we show that DROSHA protein expression in the skeletal muscle of 13-week-old *db/db* mice was decreased compared to age-matched *wt/wt* mice. From previously published data [56], we know that muscles from *db/db* mice have lower levels of *VEGF-A* mRNAs and proteins at that age. Surprisingly *miR-15a* and MDM2 protein level remained unchanged in these muscles. In these *db/db* mice, we have previously reported that insulin resistance is present in the skeletal muscle [56]. Since MDM2 activity is sensitive to insulin, it cannot be excluded that MDM2 activity is reduced in the endothelium of insulin resistant *db/db* mice without seen any changes of MDM2 protein at 13 weeks. Still, this response is remarkably different from the observations made in MECs exposed to high glucose for 6 h. Chronic exposure of the muscle tissue to high glucose over weeks might promote long-term compensatory mechanisms that significant differ from the acute response observed in MECs. It is plausible that the translational repression of *VEGF-A* by miR-15a is an early process taking place exclusively in MECs. Since miRs extraction from whole muscle retrieves miRs from all cells presents in the muscle, including the high abundant multinucleated muscle cells, it might explain why we could not detect endothelial-specific changes in miR-15a in muscle extracts. Nonetheless, the strong negative correlation observed between resting blood glucose and level of DROSHA protein in the gastrocnemius muscle of younger mice (4-week-old) further supports the notion that glucose negatively regulates DROSHA in vivo. We know that at this age, capillary rarefaction has already occurred in the muscle of *db**/db* mice [56]. Moreover, we have previously observed in young BioBreeding rats prone to develop type-1 diabetes that blood glucose levels correlated with capillary rarefaction and a reduced expression of the pro-angiogenic VEGF-A receptor 2 (VEGF-R2) [27]. The pattern observed then was similar to what was observed here in muscles of 4-week-old mice where glycemia values were negatively correlated with DROSHA protein expression.

Together, our previous and current data support the notion that DROSHA downregulation might be part of the mechanisms that promotes angiostasis during hyperglycemia in MECs. Further investigations are required to establish whether glucose-dependent downregulation of muscle DROSHA supports capillary rarefaction during the development of diabetes in vivo.

## 5. Conclusions

The present work has shown that high glucose conditions decreased the expression of DROSHA protein. Since high glucose increased the interaction between MDM2 and DROSHA and MDM2 inhibitors prevented DROSHA downregulation and miR-15a upregulation in MECs, our data suggests that a MDM2/DROSHA pathway could regulate the maturation and abundance of specific angiomiRs in these cells. Altogether our data calls for further investigations to fully understand what role the MDM2/DROSHA pathway plays in controlling the maturation of angiomiRs under hyperglycemia. Alterations of the microvasculature, particularly a reduced angiogenic capacity, are well documented in the cutaneous and skeletal muscle tissues in the context of type 1 and type 2 diabetes [25,27,28,29,30,39,83]. A thorough investigation of the role of MDM2 on DROSHA might bring new insight into the pathological process through which diabetes supports the loss of angiogenic capacity in these two tissues.

## Figures and Tables

**Figure 1 cells-10-00742-f001:**
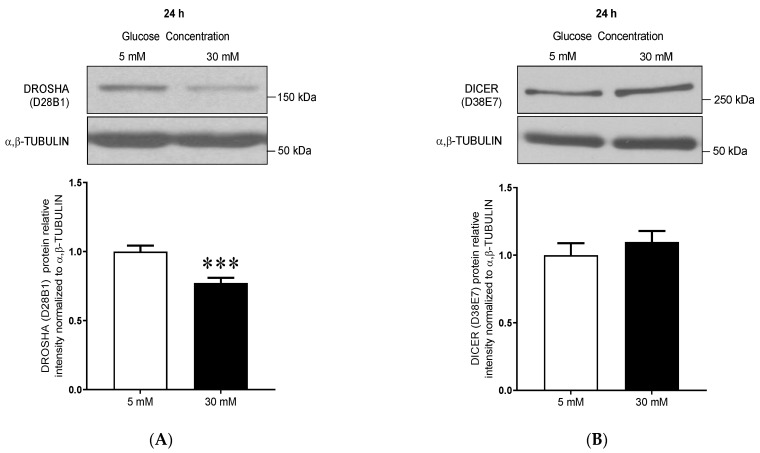
High glucose treatment decreases DROSHA protein expression in human primary dermal microvascular endothelial cells (HDMECs). (**A**,**B**) Representative immunoblots and densitometry analysis for (**A**) DROSHA (D28B1) and (**B**) DICER (D38E7) protein expression levels in primary human dermal microvascular endothelial cells (HDMEC) after 24 h treatment with low (5 mM) or high (30 mM) glucose concentrations. α/β-TUBULIN was used as a loading control. Data are means ± SEM. (*n* = 12, *t*-test, *** *p* < 0.001 vs. 5 mM). (**C**,**D**) *DROSHA* and *DICER1* mRNA expression relatively to *HPRT* in HDMEC treated with low (5 mM) or high (30 mM) glucose concentrations for 24 h. Data are means ± SEM. (*n* = 8, *DICER1* mRNA; *n* = 12, *DROSHA* mRNA).

**Figure 2 cells-10-00742-f002:**
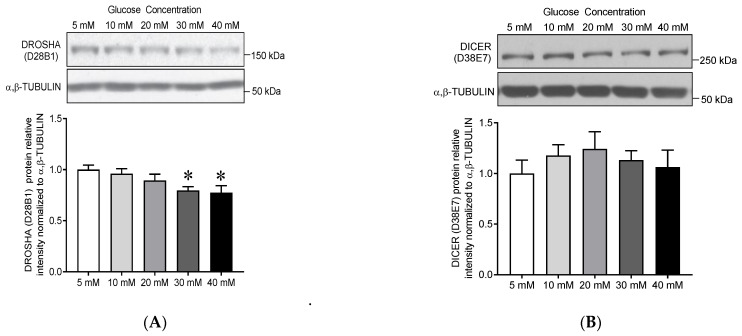
Dose and time effect of glucose treatment on DROSHA protein expression in human primary dermal microvascular endothelial cells (HDMECs). Immunoblots of (**A**) DROSHA (D28B1) and (**B**) DICER (D38E7) in human primary HDMECs treated with increasing concentrations of glucose for 24 h. α,β-TUBULIN was used as a loading control. Data are means ± SEM, *n* = 8. One-way ANOVA indicates a significant effect of glucose concentration (*p* < 0.05); * indicates significant differences compared to 5 mM glucose condition (*p* < 0.05, Bonferroni post hoc tests). (**C**) Immunoblot of DROSHA protein in HDMECs following 3, 6, and 24 h exposure to 5 and 30 mM glucose (data are means ± SEM, *n* = 3). (**D**) Immunoblots of DROSHA protein in HDMECs after 24 h treatment with 5 and 30 mM glucose, and 30 mM mannitol. Treatment with D-mannitol was used as an osmotic control. Data are means ± SEM, ** indicates significant difference compared to 30 mM glucose (** *p* < 0.01, 1-W ANOVA, Sidak Post hoc, *n* = 12).

**Figure 3 cells-10-00742-f003:**
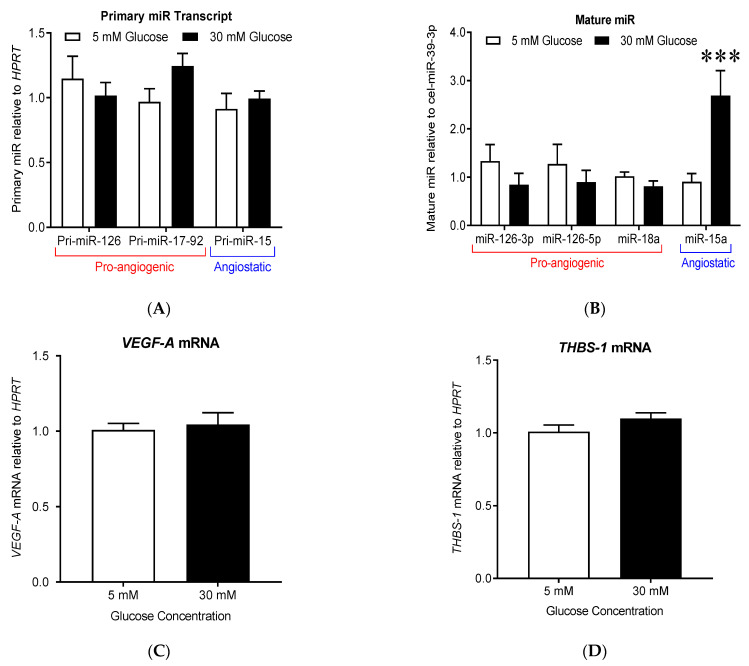
High glucose treatment shifted endothelial cell angiomiR expression towards an anti-angiogenic phenotype. (**A**) Primary miR and (**B**) mature miR levels relative to *HPRT* housekeeping gene and exogenous spike-in cel-miR-39-3p, respectively. Pro-angiogenic pri-miRs and miR are underlined in red, while angiostatic pri-miRs and miRs are underlined in blue. Data are means ± SEM (*n* = 6). Two-way ANOVA showed no effect of glucose concentration on the expression of pri-miR-126, pri-miR-17-92, and pri-miR-15; *** indicates significative difference compared to 5 mM of glucose (*p* < 0.001)). (**C**,**D**) *VEGF-A* and *THBS-1* mRNA relative to *HPRT* in HDMEC treated with 5 or 30 mM glucose concentrations for 24 h (data are means ± SEM, *n* = 9). (**E**,**F**) Immunoblots of miR targets VEGF-A and THBS-1 (A6.1) after 24 h treatment under 5 or 30 mM glucose conditions. α,β-TUBULIN was used as a loading control. Data are means ± SEM; * indicates significative differences (*p* < 0.05, *t*-test, *n* = 5-10).

**Figure 4 cells-10-00742-f004:**
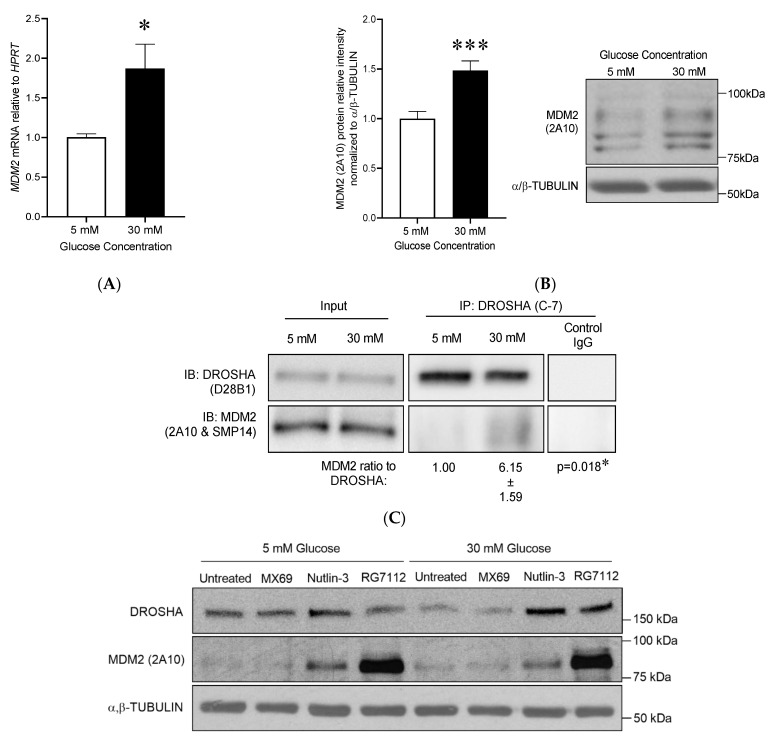
High glucose treatment increases MDM2 expression in human dermal microvascular endothelial cells (HDMECs) and enhances MDM2-DROSHA interaction. (**A**,**B**) HDMECS were treated for 24 H treatment with 30 mM or 5 mM glucose concentrations. (**A**) *MDM2* mRNA expression relative to *HPRT*. Data are means ± SEM, *n* = 9. (**B**) Immunoblot of MDM2, data are means ± SEM, *n* = 14. (**A**,**B**) * and *** indicate significant differences, *p* < 0.05 and *p* < 0.001, respectively. (**C**) After immunoprecipitation (IP) of DROSHA in glucose-treated HDMECs in presence of MG-132, the levels of MDM2 and DROSHA protein expression were measured by immunoblot analysis in whole-cell lysate (input) and in IP products (*n* = 4 per group). Normal mouse IgG was used as a control. The MDM2:DROSHA ratio is indicated as a fold of change compare to 5 mM of glucose (data are means ± SEM, *n* = 4, * *p* < 0.05). (**D**) Representative immunoblots of MDM2 and DROSHA in HDMECs treated with 30 mM or 5 mM glucose and MDM2 inhibitors (MX69, Nutlin-3 and RG7112) for 6 h. α,β-TUBULIN was used as a loading control. Quantification shows means ± SEM, *n* = 6. Two-way ANOVA shows an overall effect of MDM2 activity inhibition on DROSHA protein levels (*p* = 0.0051). Significant differences are * *p* < 0.05, *** *p* < 0.001 (Bonferroni multiple comparisons test). (**E**) Mature miR levels relative to cel-miR-39-3p in HDMECs treated with 30 mM or 5 mM glucose and Nutlin-3 for 24 H. Data are means ± SEM, *n* = 7–10. Two-ANOVA was performed for each miR. * and ** indicates significant differences *p* < 0.05 and *p* < 0.01, respectively (Bonferroni multiple comparisons test).

**Figure 5 cells-10-00742-f005:**
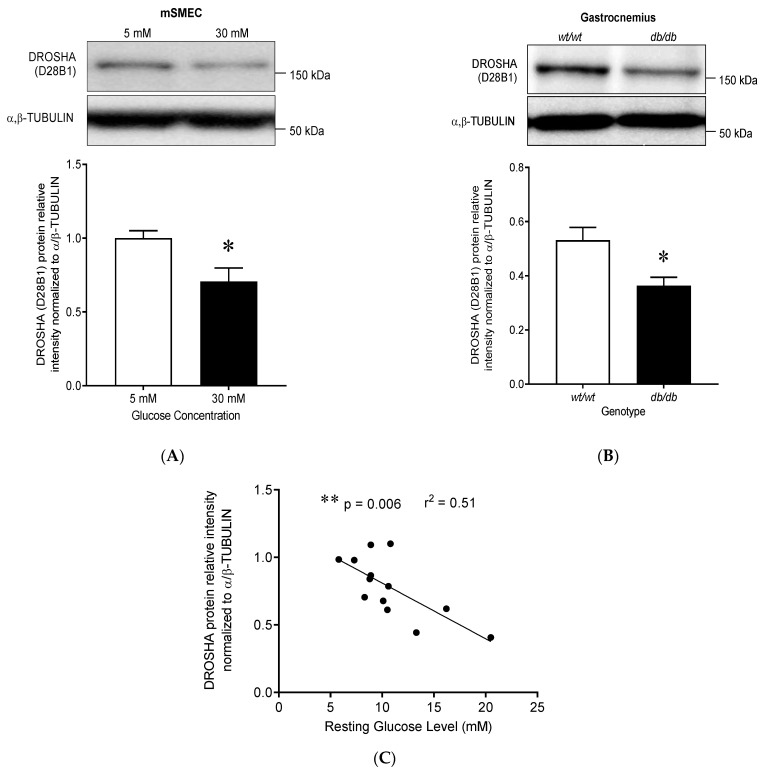
High glucose conditions decrease DROSHA protein expression in mouse skeletal muscle tissue. (**A**) Immunoblot of DROSHA expression in mouse primary endothelial cells isolated from skeletal muscle tissue (mSMECs) following 24 H treatment with high (30 mM) or low (5 mM) glucose concentrations. α/β-TUBULIN was used as a loading control. High glucose concentration (30 mM) decreased DROSHA protein expression (data are means ± SEM, *n* = 6, * *p* < 0.05). (**B**) Immunoblot of DROSHA expression in gastrocnemius muscles from wild-type (*wt/wt*) and leptin receptor deficient (*db/db*) mice (data are means ± SEM, *n* = 4–7, *p* = 0.034). The db/db mice expressed significantly less DROSHA protein compared to their wild-type littermates (data are means ± SEM, *n* = 4–7, * *p* < 0.05). (**C**) Correlation analysis between DROSHA protein expression and resting blood glucose levels in gastrocnemius muscles from 4-week-old mice, including 2 *wt/wt*, 2 *db/db* and 9 *db/wt* mice. Resting blood glucose is negatively correlated with DROSHA protein expression in mice skeletal muscle (*n* = 13, *** p* = 0.006).

**Table 1 cells-10-00742-t001:** List of Taqman probes used to measure RNA transcripts. For pri-miR and miR target name, the first three letters signify the organism: ‘hsa’ for human origin, ‘cel’ for C. elegans and ‘mmu’ for murine origin. Then, the three following letters ‘mir’ and ‘miR’ indicate sequences of pri-miR and mature miR, respectively. At the end of the target name, 3p and -5p indicate that the sequence originates from 3′ or 5′ arm, respectively.

Assay ID	Target	Sequences
Hs03303230_pri	hsa-mir-126	CGCUGGCGACGGGACAUUAUUACUUUUGGUACGC GCUGUGACACUUCAAACUCGUACCGUGAGUAAUA AUGCGCCGUCCACGGCA
Hs03302582_pri	hsa-mir-15a	CCUUGGAGUAAAGUAGCAGCACAUAAUGGUUUGU GGAUUUUGAAAAGGUGCAGGCCAUAUUGUGCUGC CUCAAAAAUACAAGG
Hs03295901_pri	hsa-mir-17	GUCAGAAUAAUGUCAAAGUGCUUACAGUGCAGGU AGUGAUAUGUGCAUCUACUGCAGUGAAGGCACUU GUAGCAUUAUGGUGAC
477887_mir	hsa-miR-126-3p	UCGUACCGUGAGUAAUAAUGCG
477888_mir	hsa-miR-126-5p	CAUUAUUACUUUUGGUACGCG
477858_mir	hsa-miR-15a-5p	UAGCAGCACAUAAUGGUUUGUG
478551_mir	hsa-miR-18a-5p	UAAGGUGCAUCUAGUGCAGAUAG
478293_mir	cel-miR-39-3p	UCACCGGGUGUAAAUCAGCUUG
482962_mir	mmu-miR-15a-5p	UAGCAGCACAUAAUGGUUUGUG
Hs02800695_m1	HPRT	GGACTAATTATGGACAGGACTGAAC
Hs01066930_m1	MDM2	GAACAAGAGACCCTGGTTAGACCAA
Hs00900055_m1	VEGF-A	ACATCACCATGCAGATTATGCGGAT
Hs00962908_m1	THBS-1	ACACAATCCGGATCAGCTGGACTCT
Hs00229023_m1	DICER1	GTGCTACCCAAAAGCAATTCCAGAG
Hs00203008_m1	DROSHA	AATGATCCGGACCTGCGCGAAGTCT

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
