# Peer review of "High Glucose Treatment Limits Drosha Protein Expression and Alters AngiomiR Maturation in Microvascular Primary Endothelial Cells via an Mdm2-dependent Mechanism"

_cells, 2021, doi:10.3390/cells10040742_

Round 1

Reviewer 1 Report

The manuscript from B. Lam et al demonstrates that high glucose reduces DROSHA expression and increases miR-15a, an anti-angiogenic miRNA. Although reduction of DROSHA expression would be expected to reduce mature miRNA levels, miR-15a specifically is increased. The authors find that MDM2 expression is increased and is binding to DROSHA under high glucose condition, explaining the reduction in DROSHA. 

The authors then inhibit MDM2 activity with Nutlin-3, and find that DROSHA expression is increased, and miR-15a expression is reduced back to normal levels. 

This is intriguing work, generating new knowledge as to how high glucose may reduce angiogenesis. 

However, the link between DROSHA and miR-15a could be clarified. Perhaps the increase in miR-15a is unrelated to DROSHA levels. siRNA knockdown of DROSHA could assess whether this is the case. Furthermore, miR-16-1 is adjacent to miR-15a in the genome. Does miR16-1 also increase under the same conditions as miR-15a?

Also, an increase in MDM2 expression would decrease p53 expression (p53 can regulate miRNAs: J Mol Cell Biol. 2011 Feb; 3(1): 44–50.). Could a decrease in p53 cause the change in expression of miR-15a? An siRNA mediated p53 knockdown experiment could address this question. 

It wold also be helpful to see whether DROSHA expression is reduced in endothelial cells from db/db mice, perhaps with immunofluorescence experiments of tissue sections.

Author Response

Please see our reply in the attached document. Thanks.

Reviewer 2 Report

The present study was designed to determine whether high glucose alters angioMiRs biogenesis through Mouse double minute 2 homolog (MDM2) dependent mechanism in human dermal microvascular endothelial cells (HDMECs). High glucose treatment increased MDM2 and miR-15s which caused downregulation of DROSHA and VEGF-A protein expressions. Inhibition of MDM2 prevented upregulation of miR-15s and downregulation of DROSHA in HDMECs. Furthermore, the authors detected downregulation of DROSHA in skeletal muscles of diabetic db/db mice.

Major concerns:

1. While most of the mechanistic studies were performed in-vitro in cell culture, it is unclear whether MDM2 and miR-15s are also increased in skeletal muscles of diabetic db/db mice. It is also unclear whether inhibition of MDM2 can prevent downregulation of DROSHA in-vivo in diabetic db/db mice. The authors need to perform additional in-vivo studies in diabetic db/db mice.

2. VEGF Antibody sc-507 of Santa Cruz has been discontinued. The authors need to provide different antibody and repeat the Western blot studies (Figure 3e). In addition, Western blots should be run with a positive control to identify a specific band.

3. Western blot of MDM2 shows several bands (Figure 4b). It is unclear which band the authors have used for quantification? Again, Western blots should be run with a positive control to identify a specific band.

4. Supplemental Data for Figure 4D: The authors took control and high glucose data from different Western blots (Figure 2D) and used as “untreated” for statistical analysis in Figure 4d. This is inappropriate because in the instructions to the authors state that “All experimental samples and controls used for one comparative analysis should be run on the same blot image. The different images should not be spliced together to illustrate the results.” Therefore, the authors need to repeat Western blots to ensure that both untreated and treated cells are loaded on the same blot.

5. The authors need to include a schematic diagram that represents the mechanisms of DROSHA downregulation.

Minor concerns:

1. Page 12, 3.4.3.: The authors made a statement that “There was an overall effect of MDM2 inhibition (P=0.0051, Figure 4d).” However, there is no bar graph for MDM2 data in Figure 4d. This should be included in the manuscript.

2. The data of blood glucose levels in db/db mice are missing.

3. The authors need to make a statement about VEGF-A, i.e. activates in endothelial cells to stimulate blood-vessel formation.

4. Western blot data in supplemental Figure 2d: How the authors have tested statistically that one outliner (1.873) can be excluded from the analysis?

5. Supplemental Figure 1b: The band pair #8 of DROSHA protein does not correspond with the data as shown in Table. This blot shows no difference; however the table shows the opposite. Explanation?

6. Figure 5 legend: "P value of 4-7", what does it mean?

Round 2

Reviewer 2 Report

Thank you for the detailed explanations based on your experiences. I have no further comments.

Author Response

Thanks. We truly appreciated your feedback and the in-depth review that you made of our work. It definitely made the manuscript stronger by providing more validation of the methods you have used. And it also helped ensuring we provided all the appropriate supplemental material.